# GENERALIZING SKILLS WITH SEMI-SUPERVISED REINFORCEMENT LEARNING

**Chelsea Finn†, Tianhe Yu†, Justin Fu†, Pieter Abbeel†‡, Sergey Levine†**
† Berkeley AI Research (BAIR), University of California, Berkeley
‡ OpenAI
{cbfinn,tianhe.yu,justinfu,pabbeel,svlevine}@berkeley.edu

## ABSTRACT

Deep reinforcement learning (RL) can acquire complex behaviors from low-level inputs, such as images. However, real-world applications of such methods require generalizing to the vast variability of the real world. Deep networks are known to achieve remarkable generalization when provided with massive amounts of labeled data, but can we provide this breadth of experience to an RL agent, such as a robot? The robot might continuously learn as it explores the world around it, even while it is deployed and performing useful tasks. However, this learning requires access to a reward function, to tell the agent whether it is succeeding or failing at its task. Such reward functions are often hard to measure in the real world, especially in domains such as robotics and dialog systems, where the reward could depend on the unknown positions of objects or the emotional state of the user. On the other hand, it is often quite practical to provide the agent with reward functions in a limited set of situations, such as when a human supervisor is present, or in a controlled laboratory setting. Can we make use of this limited supervision, and still benefit from the breadth of experience an agent might collect in the unstructured real world? In this paper, we formalize this problem setting as semi-supervised reinforcement learning (SSRL), where the reward function can only be evaluated in a set of "labeled" MDPs, and the agent must generalize its behavior to the wide range of states it might encounter in a set of "unlabeled" MDPs, by using experience from both settings. Our proposed method infers the task objective in the unlabeled MDPs through an algorithm that resembles inverse RL, using the agent's own prior experience in the labeled MDPs as a kind of demonstration of optimal behavior. We evaluate our method on challenging, continuous control tasks that require control directly from images, and show that our approach can improve the generalization of a learned deep neural network policy by using experience for which no reward function is available. We also show that our method outperforms direct supervised learning of the reward.

## 1 INTRODUCTION

Reinforcement learning (RL) provides a powerful framework for learning behavior from high-level goals. RL has been combined with deep networks to learn policies for problems such as Atari games (Mnih et al., 2015), simple Minecraft tasks (Oh et al., 2016), and simulated locomotion (Schulman et al., 2015). To apply reinforcement learning (RL) to real-world scenarios, however, the learned policy must be able to handle the variability of the real-world and generalize to scenarios that it has not seen previously. In many such domains, such as robotics and dialog systems, the variability of the real-world poses a significant challenge. Methods for training deep, flexible models combined with massive amounts of labeled data are known to enable wide generalization for supervised learning tasks (Russakovsky et al., 2015). Lifelong learning aims to address this data challenge in the context of RL by enabling the agent to continuously learn as it collects new experiences "on the job," directly in the real world (Thrun & Mitchell, 1995). However, this learning requires access to a reward function, to tell the agent whether it is succeeding or failing at its task. Although the reward is a high-level supervision signal that is in principle easier to provide than detailed labels, in practice it often depends on information that is extrinsic to the agent and is therefore difficult to measure in the real world. For example, in robotics, the reward may depend on the poses of all

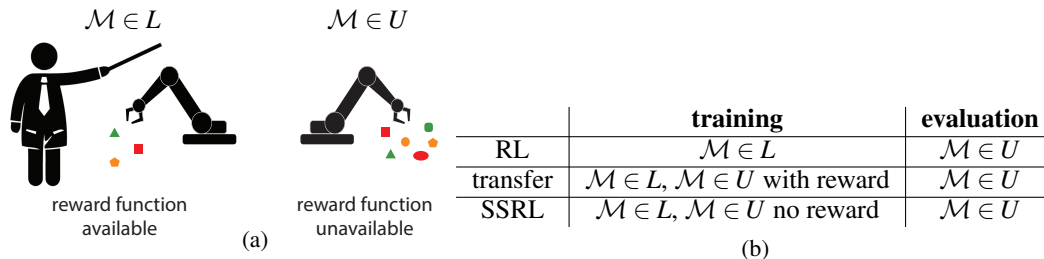

| | | training | evaluation |
|---|---|---|---|
| RL | | $\mathcal{M} \in L$ | $\mathcal{M} \in U$ |
| transfer | | $\mathcal{M} \in L$, $\mathcal{M} \in U$ with reward | $\mathcal{M} \in U$ |
| SSRL | | $\mathcal{M} \in L$, $\mathcal{M} \in U$ no reward | $\mathcal{M} \in U$ |

(b)

Figure 1: We consider the problem of semi-supervised reinforcement learning, where a reward function can be evaluated in some small set of *labeled* MDPs $\mathcal{M} \in L$, but the resulting policy must be successful on a larger set of *unlabeled* MDPs $\mathcal{M} \in L$ for which the reward function is not known. In standard RL, the policy is trained only on the labeled MDPs, while in transfer learning, the policy is finetuned using a known reward function in the unlabeled MDP set. Semi-supervised RL is distinct in that it involves using experience from the unlabeled set without access to the reward function.

of the objects in the environment, and in dialog systems, the reward may depend on the happiness of the user. This reward supervision is practical to measure in a small set of instrumented training scenarios, in laboratory settings, or under the guidance of a human teacher, but quickly becomes impractical to provide continuously to a lifelong learning system, when the agent is deployed in varied and diverse real-world settings.

Conceptually, we might imagine that this challenge should not exist, since reinforcement learning should, at least in principle, be able to handle high-level delayed rewards that can always be measured. For example, a human or animal might have their reward encode some higher-level intrinsic goals such as survival, reproduction, or the absence of pain and hunger. However, most RL methods do not operate at the level of such extremely sparse and high-level rewards, and most of the successes of RL have been in domains with natural sources of detailed external feedback, such as the score in a video game. In most real-world scenarios, such a natural and convenient score typically does not exist. It therefore seems that intelligent agents in the real world should be able to cope with only partial reward supervision, and that algorithms that enable this are of both of practical and conceptual value, since they bring us closer to real-world lifelong reinforcement learning, and can help us understand adaptive intelligent systems that can learn even under limited supervisory feedback. So how can an agent continue to learn in the real world without access to a reward function?

In this work, we formalize this as the problem of *semi-supervised reinforcement learning*, where the agent must perform RL when the reward function is known in some settings, but cannot be evaluated in others. As illustrated in Figure 1, we assume that the agent can first learn in a small range of "labeled" scenarios, where the reward is available, and then experiences a wider range of "unlabeled" scenarios where it must learn to act successfully, akin to lifelong learning in the real world. This problem statement can be viewed as being analogous to the problem of semi-supervised learning, but with the additional complexity of sequential decision making. Standard approaches to RL simply learn a policy in the scenarios where a reward function is available, and hope that it generalizes to new unseen conditions. However, it should be possible to leverage unlabeled experiences to find a more general policy, and to achieve continuous improvement from lifelong real-world experience.

Our main contribution is to propose and evaluate the first algorithm for performing semi-supervised reinforcement learning, which we call semi-supervised skill generalization (S3G). Our approach can leverage unlabeled experience to learn a policy that can succeed in a wider variety of scenarios than a policy trained only with labeled experiences. In our method, we train an RL policy in settings where a reward function is available, and then run an algorithm that resembles inverse reinforcement learning, to simultaneously learn a reward and a more general policy in the wider range of unlabeled settings. Unlike traditional applications of inverse RL algorithms, we use roll-outs from the RL policy in the labeled conditions as demonstrations, rather than a human expert, making our method completely autonomous. Although our approach is compatible with any choice of reinforcement learning and inverse reinforcement learning algorithm, we use the guided cost learning method in our experimental evaluation, which allows us to evaluate on high-dimensional, continuous robotic manipulation tasks with unknown dynamics while using a relatively modest number of samples (Finn et al., 2016). We compare our method to two baselines: (a) a policy trained with RL in settings where reward labels are available (as is standard), and (b) a policy trained in the unlabeled

settings using a reward function trained to regress to available reward labels. We find that S3G recovers a policy that is substantially more effective than the prior, standard approach in a wide variety of settings, without using any additional labeled information. We also find that, by using an inverse RL objective, our method achieves superior generalization to the reward regression approach.

## 2 Related Work

Utilizing both labeled and unlabeled data is a well-known technique that can improve learning performance when data is limited (Zhu & Goldberg, 2009). These techniques are especially important in domains where large, supervised datasets are difficult to acquire, but unlabeled data is plentiful. This problem is generally known as semi-supervised learning. Methods for solving this problem often include propagating known labels to the unlabeled examples (Zhu & Ghahramani, 2002) and using regularizing side information (Szummer & Jaakkola, 2002) such as the structure of the data. Semi-supervised learning has been performed with deep models, either by blending unsupervised and supervised objectives (Rasmus et al., 2016; Zhang et al., 2016) or by using generative models, with the labels treated as missing data (Kingma et al., 2014). Semi-supervised learning is particularly relevant in robotics and control, where collecting labeled experience on real hardware is expensive. However, while semi-supervised learning has been successful in domains such as object tracking and detection (Teichman & Thrun, 2007), applications to action and control have not been applied to the objective of the task itself.

The generalization capabilities of policies learned through RL (and deep RL) has been limited, as pointed out by Oh et al. Oh et al. (2016). That is, typically the settings under which the agent is tested do not vary from those under which it was trained. We develop a method for generalizing skills to a wider range of settings using unlabeled experience. A related but orthogonal problem is transfer learning (Taylor & Stone, 2009; Barrett et al., 2010), which attempts to use prior experience in one domain to improve training performance in another. Transfer learning has been applied to RL domains for transferring information across environments (Mordatch et al., 2016; Tzeng et al., 2016), robots (Devin et al., 2016), and tasks (Konidaris & Barto, 2006; Stolle & Atkeson, 2007; Dragan et al., 2011; Parisotto et al., 2016; Rusu et al., 2016). The goal of these approaches is typically to utilize experience in a source domain to learn faster or better in the target domain. Unlike most transfer learning scenarios, we assume that supervision cannot be obtained in many scenarios. We are also not concerned with large, systematic domain shift: we assume that the labeled and unlabeled settings come from the same underlying distribution. Note, however, that the method that we develop could be used for transfer learning problems where the state and reward are consistent across domains.

To the best of our knowledge, this paper is the first to provide a practical and tractable algorithm for semi-supervised RL with large, expressive function approximators, and illustrate that such learning actually improves the generalization of the learned policy. However, the idea of semi-supervised reinforcement learning procedures has been previously discussed as a compelling research direction by Christiano (2016) and Amodei et al. (2016).

To accomplish semi-supervised reinforcement learning, we propose a method that resembles an inverse reinforcement learning (IRL) algorithm, in that it imputes the reward function in the unlabeled settings by learning from the successful trials in the labeled settings. IRL was first introduced by Ng et al. (2000) as the problem of learning reward functions from expert, human demonstrations, typically with the end goal of learning a policy that can succeed from states that are not in the set of demonstrations (Abbeel & Ng, 2004). We use IRL to infer the reward function underlying a policy previously learned in a small set of labeled scenarios, rather than using expert demonstrations. We build upon prior methods, including guided cost learning, which propose to learn a cost and a policy simultaneously (Finn et al., 2016; Ho et al., 2016). Note that the problem that we are considering is distinct from semi-supervised inverse reinforcement learning Audiffren et al. (2015), which makes use of expert and non-expert trajectories for learning. We require a reward function in some instances, rather than expert demonstrations.

## 3 Semi-Supervised Reinforcement Learning

We first define semi-supervised reinforcement learning. We would like the problem definition to be able to capture situations where supervision, via the reward function, is only available in a small set

of labeled Markov decision processes (MDPs), but where we want our agent to be able to continue to learn to perform successfully in a much larger set of unlabeled MDPs, where reward labels are unavailable. For example, if the task corresponds to an autonomous car learning to drive, the labeled MDPs might correspond to a range of closed courses, while the unlabeled MDPs might involve driving on real-world highways and city streets. We use the terms labeled and unlabeled in analogy to semi-supervised learning, but note a reward observation is not as directly informative as a label.

Formally, we consider a distribution $p(\mathcal{M})$ over undiscounted finite-horizon MDPs, each defined as a 4-tuple $\mathcal{M}_i = (S, A, T, R)$ over states, actions, transition dynamics (which are generally unknown), and reward. The states and actions may be continuous or discrete, and the reward function $R$ is assumed to the same across MDPs in the distribution $p(\mathcal{M})$. Let $L$ and $U$ denote two sets of MDPs sampled from the distribution $p(\mathcal{M})$. Experience may be collected in both sets of MDPs, but the reward can only be evaluated in the set of labeled MDPs $L$. The objective is to find a policy $\pi^*$ that maximizes expected reward in the distribution over MDPs:

$$\pi^* = \arg\max_{\pi} \ \mathbb{E}_{\pi, p(\mathcal{M})} \left[ \sum_{t=0}^{H} R(s_t, a_t) \right],$$

where $H$ denotes the horizon. Note that the notion of finding a policy that succeeds on a distribution of MDPs is very natural in many real-world reinforcement learning problems. For example, in the earlier autonomous driving example, our goal is not to find a policy that succeeds on one particular road or in one particular city, but on all roads that the car might encounter. Note that the problem can also be formalized in terms of a single large MDP with a large diversity of initial states, but viewing the expectation as being over a distribution of MDPs provides a more natural analogue with semi-supervised learning, as we discuss below.

In standard semi-supervised learning, it is assumed that the data distribution is the same across both labeled and unlabeled examples, and the amount of labeled data is limited. Similarly, semi-supervised reinforcement learning assumes that the labeled and unlabeled MDPs are sampled from the same distribution. In SSRL, however, it is the set of labeled MDPs that is limited, whereas acquiring large amounts of experience within the set of labeled MDPs is permissible, though unlimited experience in the labeled MDPs is not sufficient on its own for good performance on the entire MDP distribution. This is motivated by real-world lifelong learning, where an agent (e.g. a robot) may be initially trained with detailed reward information in a small set of scenarios (e.g. with a human teacher), and is then deployed into a much larger set of scenarios, without reward labels. One natural question is how much variation can exist in the distribution over MDPs. We empirically answer this question in our experimental evaluation in Section 5.

The standard paradigm in reinforcement learning is to learn a policy in the labeled MDPs and apply it directly to new MDPs from the same distribution, hoping that the original policy will generalize (Oh et al., 2016). An alternative approach is to train a reward function with supervised learning to regress from the agent's observations to the reward labels, and then use this reward function for learning in the unlabeled settings. In our experiments, we find that this approach is often more effective because, unlike the policy, the reward function is decoupled from the rest of the MDP, and can thus generalize more readily. The agent can then continue to learn from unlabeled experiences using the learned reward function. However, because the state distributions in the two sets of MDPs may be different, a function approximator trained on the reward function in the labeled MDPs may not necessarily generalize well to the unlabeled one, due to the domain shift. A more effective solution would be to incorporate the unlabeled experience sampled from $U$ when learning the reward. Unlike typical semi-supervised learning, the goal is not to learn the reward labels per se, but to learn a policy that optimizes the reward. By incorporating both labeled and unlabeled experience, we can develop an algorithm that alternates between inferring the reward function and updating the policy, which effectively provides a shaping, or curriculum, for learning to perform well in the unlabeled settings. In the following section, we discuss our proposed algorithm in detail.

## 4 Semi-Supervised Skill Generalization

We now present our approach for performing semi-supervised reinforcement learning for generalizing previously learned skills. As discussed previously, our goal is to learn a policy that maximizes expected reward in $\mathcal{M} \in U$, using both unlabeled experience in $U$ and labeled experience in $L$. We

will use the formalism adopted in the previous section; however, note that performing RL in a set of MDPs can be equivalently be viewed as a single MDP with a large diversity of initial conditions.

In order to perform semi-supervised reinforcement learning, we use the framework of maximum entropy control (Ziebart, 2010; Kappen et al., 2012), also called linear-solvable MDPs (Dvijotham & Todorov, 2010). This framework is a generalization of the standard reinforcement learning formulation, where instead of optimizing the expected reward, we optimize an entropy-regularized objective of the form

$$\pi_{\text{RL}} = \arg\max_{\pi} \mathbb{E}_{\pi,\mathcal{M}\in L}\left[\sum_{t=0}^{H} R(s_t, a_t)\right] - \mathcal{H}(\pi). \tag{1}$$

To see that this is a generalization of the standard RL setting, observe that, as the magnitude of the reward increases, the relative weight on the entropy regularizer decreases, so the classic RL objective can be recovered by putting a temperature $\beta$ on the reward, and taking the limit as $\beta \to \infty$. For finite rewards, this objective encourages policies to take random actions when all options have roughly equal value. Under the optimal policy $\pi_{\text{RL}}$, samples with the highest reward $R$ have the highest likelihood, and the likelihood decreases exponentially with decrease in reward. In our work, this framework helps to produce policies in the labeled MDP that are diverse, and therefore better suited for inferring reward functions that transfer effectively to the unlabeled MDP.

After training $\pi_{\text{RL}}$, we generate a set of samples from $\pi_{\text{RL}}$ in $L$, which we denote as $\mathcal{D}_{\pi_{\text{RL}}}$. The objective of S3G is to use $\mathcal{D}_{\pi_{\text{RL}}}$ to find a policy that maximizes expected reward in $U$,

$$\max_{\theta}\ \mathbb{E}_{\pi_\theta,\mathcal{M}\in U}\left[\sum_{t=0}^{T} R(s_t, a_t)\right] - \mathcal{H}(\pi_\theta),$$

where the reward $R$ is not available. By using the agent's prior experience $\mathcal{D}_{\pi_{\text{RL}}}$, as well as unlabeled experience in $U$, we aim to learn a well-shaped reward function to facilitate learning in $U$. To do so, S3G simultaneously learns a reward function $\tilde{R}_\phi$ with parameters $\phi$ and optimizes a policy $\pi_\theta$ with parameters $\theta$ in the unlabeled MDP $U$. This consists of iteratively taking samples $\mathcal{D}_{\pi_\theta}$ from the current policy $\pi_\theta$ in $U$, updating the reward $\tilde{R}_\phi$, and updating the policy $\pi$ using reward values imputed using $\tilde{R}_\phi$. At the end of the procedure, we end up with a policy $\pi_\theta$ optimized in $U$. As shown in prior work, this procedure corresponds to an inverse reinforcement learning algorithm that converges to a policy that matches the performance observed in $\mathcal{D}_{\pi_{\text{RL}}}$ (Finn et al., 2016). We next go over the objectives used for updating the reward and the policy.

**Reward update:** Because of the entropy regularized objective in Equation 1, it follows that the samples $\mathcal{D}_{\pi_{\text{RL}}}$ are generated from the following maximum entropy distribution (Ziebart, 2010):

$$p(\tau) = \frac{1}{Z}\exp(R(\tau)), \tag{2}$$

where $\tau$ denotes a single trajectory sample $\{s_0, a_0, s_1, a_1, ..., s_T\}$ and $R(\tau) = \sum_t R(s_t, a_t)$. Thus, the objective of the reward optimization phase is to maximize the log likelihood of the agent's prior experience $\mathcal{D}_{\pi_{\text{RL}}}$ under this exponential model. The computational challenge here is to estimate the partition function $Z$ which is intractable to compute in high-dimensional spaces. We thus use importance sampling, using samples to estimate the partition function $Z$ as follows:

$$\mathcal{L}(\phi) = \sum_{\tau \sim \mathcal{D}_{\pi_{\text{RL}}}} \tilde{R}_\phi(\tau) - \log Z \ \approx \ \sum_{\tau \sim \mathcal{D}_{\pi_{\text{RL}}}} \tilde{R}_\phi(\tau) - \log \sum_{\tau \sim \mathcal{D}_{\text{samp}}} \frac{\exp(\tilde{R}_\phi(\tau))}{q(\tau)}, \tag{3}$$

where $\mathcal{D}_{\text{samp}}$ is the set of samples used for estimating the partition function $Z$ and $q(\tau)$ is the probability of sampling $\tau$ under the policy it was generated from. Note that the distribution of this set of samples is crucial for effectively estimating $Z$. The optimal distribution for importance sampling is the one that is proportional to $q(\tau) \propto |\exp(\tilde{R}_\phi(\tau))| = \exp(\tilde{R}_\phi(\tau))$. Conveniently, this is also the optimal behavior when the reward function is fully optimized such that $\tilde{R}_\phi \approx R$. Thus, we adaptively update the policy to minimize the KL-divergence between its own distribution and the distribution induced by the current reward, $\tilde{R}_\phi(\tau)$, and use samples from the policy to estimate the partition function. Since the importance sampling estimate of $Z$ will be high variance at the beginning of training when fewer policy samples have been collected, we also use the samples from the RL policy $\pi_{\text{RL}}$. Thus we set $\mathcal{D}_{\text{samp}}$ to be $\{\mathcal{D}_{\pi_\theta}\bigcup\mathcal{D}_{\pi_{\text{RL}}}\}$.

---
**Algorithm 1** Semi-Supervised Skill Generalization

---
0: **inputs:** Set of unlabeled MDPs $U$; reward $R$ for labeled MDPs $\mathcal{M} \in L$
1: Optimize $\pi_{\text{RL}}$ to maximize $R$ in $\mathcal{M} \in L$
2: Generate samples $\mathcal{D}_{\pi_{\text{RL}}}$ from $\pi_{\text{RL}}$ in $\mathcal{M} \in L$
3: Initialize $\mathcal{D}_{\text{samp}} \leftarrow \mathcal{D}_{\pi_{\text{RL}}}$
4: **for** iteration $i = 1$ to $I$ **do**
5:     Run $\pi_\theta$ in $\mathcal{M} \in U$ to generate samples $\mathcal{D}_{\pi_\theta}$
6:     Append samples $\mathcal{D}_{\text{samp}} \leftarrow \mathcal{D}_{\text{samp}} \cup \mathcal{D}_{\pi_\theta}$
7:     Update reward $\tilde{R}_\phi$ according to Equation 3 using $\mathcal{D}_{\pi_{\text{RL}}}$ and $\mathcal{D}_{\text{samp}}$
8:     Update policy $\pi_\theta$ according to Equation 4, using $\tilde{R}_\phi$ and $\mathcal{D}_{\pi_\theta}$
9: **end for**
10: **return** generalized policy $\pi_\theta$

---

We parameterize the reward using a neural network, and update it using mini-batch stochastic gradient descent, by backpropagating the gradient of the Equation 3 to the parameters of the reward.

**Policy update:**   Our goal with the policy is two-fold. First, we of course need a policy that succeeds in MDPs $\mathcal{M} \in U$. But since the reward in these MDPs is unavailable, the policy must also serve to generate samples for more accurately estimating the partition function in Equation 2, so that the reward update step can improve the accuracy of the estimated reward function. The policy optimization objective to achieve both of these is to maximize the expected reward $\tilde{R}_\phi$, augmented with an entropy term as before:

$$\mathcal{L}(\theta) = \ \mathbb{E}_{\pi_\theta, \mathcal{M} \in U}\left[\sum_{t=0}^{T} \tilde{R}_\phi(s_t, a_t)\right] - \mathcal{H}(\pi_\theta) \tag{4}$$

While we could in principle use any policy optimization method in this step, our prototype uses mirror descent guided policy search (MDGPS), a sample-efficient policy optimization method suitable for training complex neural network policies that has been validated on real-world physical robots (Montgomery & Levine, 2016; Montgomery et al., 2016). We interleave reward function updates using the objective in Equation 3 within the policy optimization method. We describe the policy optimization procedure in detail in Appendix A.

The full algorithm is presented in Algorithm 1. Note that this iterative procedure of comparing the current policy to the optimal behavior provides a form of shaping or curriculum to learning. Our method is structured similarly to the recently proposed guided cost learning method (Finn et al., 2016), and inherits its convergence properties and theoretical foundations. Guided cost learning is an inverse RL algorithm that interleaves policy learning and reward learning directly in the target domain, which in our case is the unlabeled MDPs. Unlike guided cost learning, however, the cost (or reward) is not inferred from expert human-provided demonstrations, but from the agent's own prior experience in the labeled MDPs.

## 5  EXPERIMENTAL EVALUATION

Since the aim of S3G is to improve the generalization performance of a learned policy by leveraging data from the unlabeled MDPs, our experiments focus on domains where generalization is critical for success. Despite the focus on generalization in many machine learning problems, the generalization capabilities of policies trained with RL have frequently been overlooked. For example, in recent RL benchmarks such as the Arcade Learning Environment (Bellemare et al., 2012) and OpenAI Gym (Brockman et al., 2016), the training conditions perfectly match the testing conditions. Thus, we define our own set of simulated control tasks for this paper, explicitly considering the types of variation that a robot might encounter in the real world. Through our evaluation, we seek to measure how well semi-supervised methods can leverage unlabeled experiences to improve the generalization of a deep neural network policy learned only in only labeled scenarios.

Code for reproducing the simulated experiments is available online[1]. Videos of the learned policies can be viewed at `sites.google.com/site/semisupervisedrl`.

---
[1]The code is available at `github.com/cbfinn/gps/tree/ssrl`

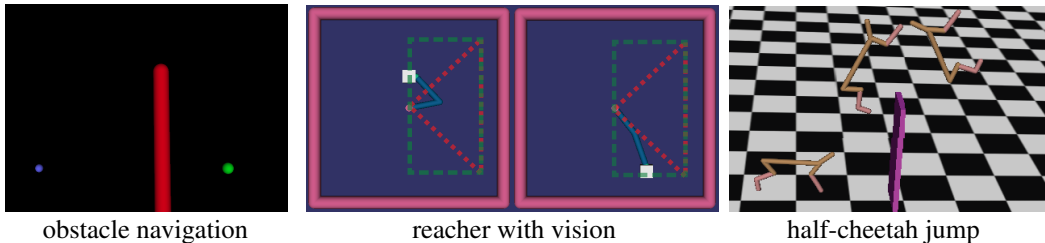

obstacle navigation                    reacher with vision                    half-cheetah jump

Figure 2: Illustrations of the tasks. For the reacher with vision, the range of the target for the labeled MDPs is shown with a red dotted line, and for the unlabeled MDPs with a green dashed line. For the obstacle and cheetah tasks, we show the highest obstacle height.

## 5.1 TASKS

Each of the tasks are modeled using the MuJoCo simulator, and involve continuous state and action spaces with unknown dynamics. The task difficulty ranges from simple, low-dimensional problems to tasks with complex dynamics and high-dimensional observations. In each experiment, the reward function is available in some settings but not others, and the unlabeled MDPs generally involve a wider variety of conditions. We visualize the tasks in Figure 2 and describe them in detail below:

**obstacle navigation / obstacle height:** The goal of this task is to navigate a point robot around an obstacle to a goal position in 2D. The observation is the robot's position and velocity, and does not include the height of the obstacle. The height of the obstacle is 0.2 in the labeled MDP, and 0.5 in the unlabeled MDP.

**2-link reacher / mass:** This task involves moving the end-effector of a two-link reacher to a specified goal position. The observation is the robot's joint angles, end-effector pose, and their time-derivatives. In the labeled MDPs, the mass of the arm varies between $7 \times 10^{-9}$ and $7 \times 10^{1}$, whereas the unlabeled MDPs involve a range of $7 \times 10^{-9}$ to $7 \times 10^{3}$.

**2-link reacher with vision / target position:** The task objective is the same as the 2-link reacher, except, in this task, the MDPs involve a wide 2D range of target positions, shown in Figure 2. Instead of passing in the coordinate of the target position, the policy and the reward function receive a raw $64 \times 80$ RGB image of the environment at the first time step.

**half-cheetah jump / wall height:** In this task, the goal is for a simulated 6-DOF cheetah-like robot with to jump over a wall, with 10% gravity. The observation is the robot's joint angles, global pose, and their velocities, for a total dimension of 20. The unlabeled MDP involves jumping over a 0.5 meter wall, compared to the labeled MDP with a 0.2 meter wall. Success is measured based on whether or not the cheetah fully clears the wall. Policies for reward regression, S3G, and oracle were initialized from the RL policy.

In all tasks, the continuous action vector corresponds to the torques or forces applied to each of the robot's joints. For the first three tasks, reaching the goal position within 5 cm is considered a success. For the non-visual tasks, the policy was represented using a neural network with 2 hidden layers of 40 units each. The vision task used 3 convolutional layers with 15 filters of size $5 \times 5$ each, followed by the spatial feature point transformation proposed by Levine et al. (2016), and lastly 3 fully-connected layers of 20 units each. The reward function architecture mirrored the architecture as the policy, but using a quadratic norm on the output, as done by Finn et al. (2016).

## 5.2 EVALUATION

In our evaluation, we compare the performance of S3G to that of (i) the RL policy $\pi_{RL}$, trained only in the labeled MDPs, (ii) a policy learned using a reward function fitted with supervised learning, and (iii) an oracle policy which can access the true reward function in all scenarios. The architecture of the reward function fitted with supervised learning is the same as that used in S3G.

To extensively test the generalization capabilities of the policies learned with each method, we measure performance on a wide range of settings that is a superset of the unlabeled and labeled MDPs, as indicated in Figure 3. We report the success rate of policies learned with each method in Table 1,

Table 1: The success rate of each method with respect to generalization. The table compares the standard RL policy (which is trained only on the labeled MDPs), with both the supervised regression method and S3G. Both of the latter use the unlabeled regime for additional training, though only S3G also uses the unlabeled data to improve the learned reward function.

|  | RL policy | reward regression (ours) | S3G (ours) | oracle |
|---|---|---|---|---|
| obstacle | 65% | 29% | **79%** | 36% |
| 2-link reacher | 75% | 60% | **98%** | 80% |
| 2-link reacher with vision | 69% | 85% | **92%** | 100% |
| half-cheetah | 56% | 73% | **79%** | 86% |

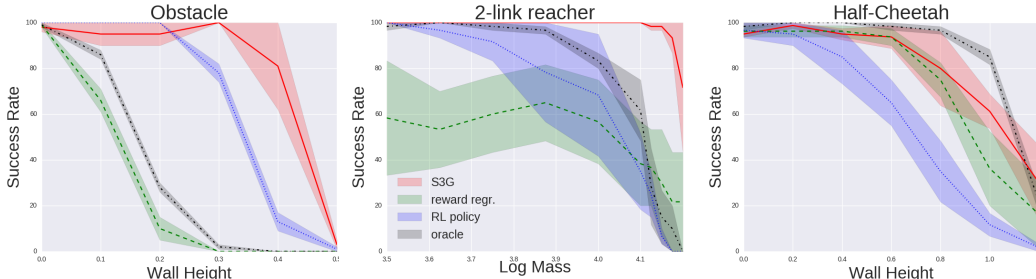

Figure 3: Generalization capability of the obstacle, 2-link reacher, and half-cheetah tasks as a function of the task variation. Performance for these tasks is averaged over 3 random seeds.

and visualize the generalization performance in the 2-link reacher, cheetah, and obstacle tasks in Figure 3. The sample complexity of each method is reported in Appendix B.

In all four tasks, the RL policy $\pi_{RL}$ generalizes worse than S3G, which demonstrates that, by using unlabeled experience, we can indeed improve generalization to different masses, target positions, and obstacle sizes. In the obstacle and both reacher tasks, S3G also outperforms reward regression, suggesting that it is also useful to use unlabeled experience to learn the reward.

In the obstacle task, the results demonstrate that the reward functions learned using S3G actually produce better generalization in some cases than learning on both the labeled and unlabeled MDPs with full knowledge of the true reward function. While this may at first seem counterintuitive, this agrees with the observation in prior work Guo et al. (2013) that the true reward function is not always the best one when learning with limited samples, computational power, or representational capacity (i.e. because it is not sufficiently shaped). S3G also outperforms the oracle and reward regression in the 2-link reacher task, indicating that the learned reward shaping is also beneficial in that task.

For the vision task, the visual features learned via RL in the labeled MDPs were used to initialize the vision layers of the reward and policy. We trained the vision-based reacher with S3G with both end-to-end finetuning of the visual features and with the visual features frozen and only the fully-connected layers trained on the unlabeled MDPs. We found performance to be similar in both cases, suggesting that the visual features learned with RL were good enough, though fine-tuning the features end-to-end with the inverse RL objective did not hurt the performance.

## 6    CONCLUSION & FUTURE WORK

We presented the first method for semi-supervised reinforcement learning, motivated by real-world lifelong learning. By inferring the reward in settings where one is not available, S3G can improve the generalization of a learned neural network policy trained only in the "labeled" settings. Additionally, we find that, compared to using supervised regression to reward labels, we can achieve higher performance using an inverse RL objective for inferring the reward underlying the agent's prior experience. Interestingly, this does not directly make use of the reward labels when inferring the reward of states in the unlabeled MDPs, and our results on the obstacle navigation task in fact suggest that the rewards learned with S3G exhibit better shaping.

As we discuss previously, the reward and policy optimization methods that we build on in this work are efficient enough to learn complex tasks with hundreds of trials, making them well suited for

learning on physical systems such as robots. Indeed, previous work has evaluated similar methods on real physical systems, in the context of inverse RL (Finn et al., 2016) and vision-based policy learning (Levine et al., 2016). Thus, it is likely feasible to apply this method for semi-supervised reinforcement learning on a real robotic system. Applying S3G on physical systems has the potential to enable real-world lifelong learning, where an agent is initialized using a moderate amount of labeled experience in a constrained setting, such as a robot learning a skill for the first time in the lab, and then allowed to explore the real world while continuous improving its capabilities without additional supervision. This type of continuous semi-supervised reinforcement learning has the potential to remove the traditional distinction between a training and test phase for reinforcement learning agents, providing us with autonomous systems that continue to get better with use.

## ACKNOWLEDGMENTS

The authors would like to thank Anca Dragan for insightful discussions, and Aviv Tamar and Roberto Calandra for helpful feedback on the paper. Funding was provided by the NSF GRFP, the DARPA Simplex program, and Berkeley DeepDrive.

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

## A    MIRROR DESCENT GUIDED POLICY SEARCH

To optimize policies with S3G, we chose to use mirror-descent guided policy search (MDGPS), for its superior sample efficiency over other policy optimization methods. MDGPS belongs to a class of guided policy search methods, which simplify policy search by decomposing the problem into two phases: a) a trajectory-centric RL phase (C-phase) and b) a supervised learning phase (S-phase). During the C-phase, a trajectory-centric RL method is used to train "local" controllers for each of M initial positions. In the S-phase, a global policy $\pi_\theta(a|s)$ is trained using supervised learning to match the output of each of the local policies.

MDGPS can be interpreted as an approximate variant of mirror-descent on the expected cost $J(\theta) = \sum_{t=1}^{T} \mathbb{E}_{\pi_\theta(s_t,a_t)}[-R(s_t,a_t)]$ under policy's trajectory distribution, where $\pi_\theta(s_t,a_t)$ denotes the marginal of $\pi_\theta(\tau) = p(s_1)\prod_{t=1}^{T} p(s_{t+1}|s_t,a_t)\pi(a_t|s_t)$ and $\tau = \{s_1,a_1,\ldots,s_T,a_T\}$ denotes the trajectory. In the C-phase, we learn new local policies for each initial position, and in the S-phase we project the local policies down to a single global policy $\pi_\theta$, using KL divergence as the distance metric.

To produce local policies, we make use of the iterative linear quadratic regulator (iLQR) algorithm to train time-varying linear-Gaussian controllers. iLQR makes up for its weak representational power by being sample efficient under regimes where it is capable of learning. Usage of iLQR requires a twice-differentiable cost function and linearized dynamics.

In order to fit a dynamics model, we use the recent samples to fit a gaussian mixture model (GMM) on $(s_t,a_t,s_{t+1})$ tuples. We then use linear regression to fit time-varying linear dynamics of the form $s_{t+1} = F_t s_t + f_t$ on local policy samples from the most recent iteration, using the clusters from the GMM as a normal-inverse Wishart prior.

During the C-step, for each initial condition $m$, we optimize the entropy-augmented of the form, objective constrained against the global policy:

$$q_m = \arg\max_q \mathbb{E}_{q,p_m(s_0)}\left[\sum_{t=0}^{T} R(s_t,a_t)\right] - \mathcal{H}(q) \text{ s.t. } \mathcal{D}_{KL}(q||\pi_\theta) \leq \varepsilon$$

Where $R(s_t,a_t)$ is a twice-differentiable objective such as $L2$-distance from a target state.

This optimization results in a local time-varying linear-Gaussian controller $q_m(\mathbf{s}_t|\mathbf{a}_t) \sim \mathcal{N}(K_{m,t}s_t + k_{m,t}, C_{m,t})$ which is executed to obtain supervised learning examples for the S-step.

## B    SAMPLE COMPLEXITY OF EXPERIMENTS

Because we use guided policy search to optimize the policy, we inherit its sample efficiency. In Table 2, we report the number of samples used in both labeled and unlabeled scenarios for all tasks and all methods. Note that the labeled samples used by the oracle are in from the "unlabeled" MDPs $U$, where we generally assume that reward labels are not available.

Table 2: Sample complexity of each experiment. This table records the total number of samples used to train policies in the labeled setting (RL and oracle), and the unlabeled setting (reward regression, S3G). The sample complexity of unlabeled experiments is denoted as (unlabeled samples + labeled samples)

|  | Labeled | | Unlabeled + Labeled | |
|---|---|---|---|---|
|  | RL | oracle | reward regression | S3G |
| obstacle | 250 | 250 | 300+250 | 300+250 |
| 2-link reacher | 200 | 300 | 900+200 | 900+200 |
| 2-link reacher with vision | 250 | 650 | 1170+250 | 1300+250 |
| half-cheetah | 600 | 600 | 1400+600 | 1400+600 |

