# Peer review of "Generalizing Skills with Semi-Supervised Reinforcement Learning"

_ICLR 2017 — accepted_

[Official Review · AnonReviewer2 · rating 8 · confidence 4 · 16 Dec 2016]

The paper proposes to study the problem of semi-supervised RL where one has to distinguish between labelled MDPs that provide rewards, and unlabelled MDPs that are not associated with any reward signal. The underlying is very simple since it aims at simultaneously learning a policy based on the REINFORCE+entropy regularization technique, and also a model of the reward that will be used (as in inverse reinforcement learning) as a feedback over unlabelled MDPs. The experiments are made on different continous domains and show interesting results

The paper is well written, and easy to understand. It is based on a simple but efficient idea of simultaneously learning the policy and a model of the reward and the resulting algorithm exhibit interesting properties. The proposed idea is quite obvious, but the authors are the first ones to propose to test such a model. The experiments could be made stronger by mixing continuous and discrete problems but are convincing.

[Official Review · AnonReviewer3 · rating 6 · confidence 5 · 17 Dec 2016]
**An approach to semi supervised RL using inverse RL**
soundness 5 · originality 5 · clarity 5 · recommendation (unofficial) 2

In supervised learning, a significant advance occurred when the framework of semi-supervised learning was  adopted, which used the weaker approach of unsupervised learning to infer some property, such as a distance measure or a smoothness regularizer, which could then be used with a small number of labeled examples. The approach rested on the assumption of smoothness on the manifold, typically. 

This paper attempts to stretch this analogy to reinforcement learning, although the analogy is somewhat incoherent. Labels are not equivalent to reward functions, and positive or negative rewards do not mean the same as positive and negative labels. Still, the paper makes a worthwhile attempt to explore this notion of semi-supervised RL, which is clearly an important area that deserves more attention. The authors use the term "labeled MDP" to mean the typical MDP framework where the reward function is unknown. They use the confusing term "unlabeled MDP" to mean the situation where the reward is unknown, which is technically not an MDP (but a controlled Markov process). 

In the classical RL transfer learning setup, the agent is attempting to transfer learning from a source "labeled" MDP to a target "labeled" MDP (where both reward functions are known, but the learned policy is known only in the source MDP). In the semi-supervised RL setting, the target is an "unlabeled" CMP, and the source is both a "labeled" MDP and an "unlabeled" CMP. The basic approach is to use inverse RL to infer the unknown "labels" and then attempt to construct transfer. A further restriction is made to linearly solvable MDPs for technical reasons. Experiments are reported using three relatively complex domains using the Mujoco physics simulator. 

The work is interesting, but in the opinion of this reviewer, the work fails to provide a simple sufficiently general notion of semi-supervised RL that will be of sufficiently wide interest to the RL community. That remains to be done by a future paper, but in the interim, the work here is sufficiently interesting and the problem is certainly a worthwhile one to study.

[Official Review · AnonReviewer4 · rating 7 · confidence 3 · 17 Dec 2016]
clarity 5

This paper formalizes the problem setting of having only a subset of available MDPs for which one has access to a reward. The authors name this setting "semi-supervised reinforcement learning" (SSRL), as a reference to semi-supervised learning (where one only has access to labels for a subset of the dataset). They provide an approach for solving SSRL named semi-supervised skill generalization (S3G), which builds on the framework of maximum entropy control. The whole approach is straightforward and amounts to an EM algorithm with partial labels (: they alternate iteratively between estimating a reward function (parametrized) and fitting a control policy using this reward function. They provide experiments on 4 tasks (obstacle, 2-link reacher, 2-link reacher with vision, half-cheetah) in MuJoCo.

The paper is well-written, and is overall clear. The appendix provides some more context, I think a few implementation details are missing to be able to fully reproduce the experiments from the paper, but they will provide the code.

The link to inverse reinforcement learning seems to be done correctly. However, there is no reference to off-policy policy learning, and, for instance, it seems to me that the \tau \in D_{samp} term of equation (3) could benefit from variance reduction as in e.g. TB(\lambda) [Precup et al. 2000] or Retrace(\lambda) [Munos et al. 2016].

The experimental section is convincing, but I would appreciate a precision (and small discussion) of this sentence "To extensively test the generalization capabilities of the policies learned with each method, we measure performance on a wide range of settings that is a superset of the unlabeled and labeled MDPs" with numbers for the different scenarios (or the replacement of superset by "union" if this is the case). It may explain better the poor results of "oracle" on "obstacle" and "2-link reacher", and reinforce* the further sentences "In the obstacle task, the true reward function is not sufficiently shaped for learning in the unlabeled MDPs. Hence, the reward regression and oracle methods perform poorly".

Correction on page 4: "5-tuple M_i = (S, A, T, R)" is a 4-tuple.

Overall, I think that this is a good and sound paper. I am personally unsure as to if all the parallels and/or references to previous work are complete, thus my confidence score of 3.

(* pun intended)

[Final Decision · Program Chairs · 06 Feb 2017]
**ICLR committee final decision**

This paper provides an interesting framework for handling semi-supervised RL problems, settings were one can interact with many MDPs drawn from some class, but where only a few have observable rewards; the agent then uses a policy derived from the labeled MDPs to estimate a reward function for the unlabeled MDPs. The approach is straightforward, and one reviewer raises the reasonable concern that this seems to be a fairly narrow definition and approach to the universe of things that could be thought of as semi-supervised reinforcement learning (elements like Laplacian value functions, e.g. proto value functions, for instance, present a very different take on could what also be considered semi-supervised RL).
 
 Overall, however, the main benefit of this paper is that the overall idea here is quite compelling. Even if it's a narrow view on semisupervised RL, it nonetheless is clearly thinking more broadly about skill and data efficiency than what is common in many current RL papers. Given this impressive scope, plus good performance (if only on a relatively small set of benchmarks), it seems like this paper is certainly above bar for acceptance.
 
 Pros:
 + Nice introduction of a new/modified semisupervised reinforce setting
 + Results on benchmarks look compelling (if still fairly small scale)
 
 Cons:
 - Rather limited view of the space of all possible semisupervised Rl
 - Results on hardest task (half-cheetah) aren't _that_ much better than much simpler approaches